# Proposed Diagnostic Criteria for Invasive Pulmonary Aspergillosis in Patients with Autoimmune Inflammatory Rheumatic Diseases: A Proof-of-Concept Study [note 1]

**DOI:** 10.3390/jof11060437

**Published:** 2025-06-07

**Authors:** Takashi Kurita, Koh Okamoto, Noritaka Sekiya, Ryoichi Hanazawa, Akio Yamamoto, Tadashi Hosoya, Akihiro Hirakawa, Shinsuke Yasuda, Yoshiaki Gu

**Affiliations:** 1Department of Infectious Diseases, Institute of Science Tokyo, Tokyo 113-8510, Japan; kohokamoto.cid@tmd.ac.jp (K.O.); sekiya.noritaka@tmd.ac.jp (N.S.); yogu.cid@tmd.ac.jp (Y.G.); 2Center for Infectious Disease Education and Analysis (TCIDEA), Institute of Science Tokyo, Tokyo 113-8510, Japan; 3Department of Infectious Disease Emergency Preparedness, Institute of Science Tokyo, Tokyo 113-8510, Japan; 4Department of Clinical Biostatistics, Institute of Science Tokyo, Tokyo 113-8510, Japan; r-hanazawa.crc@tmd.ac.jp (R.H.); a-hirakawa.crc@tmd.ac.jp (A.H.); 5Department of Rheumatology, Institute of Science Tokyo, Tokyo 113-8510, Japan; yamamoto.rheu@tmd.ac.jp (A.Y.); hosorheu@tmd.ac.jp (T.H.); syasuda.rheu@tmd.ac.jp (S.Y.)

**Keywords:** aspergillosis, autoimmune diseases, diagnosis, immunosuppressed host, invasive pulmonary aspergillosis

## Abstract

The EORTC/MSGERC definition lacks sufficient sensitivity for diagnosing invasive pulmonary aspergillosis (IPA) in patients with autoimmune inflammatory rheumatic diseases (AIIRDs). We hypothesized that the partial fulfillment of the EORTC/MSGERC definition can improve its diagnostic sensitivity. This retrospective observational study included patients with AIIRDs on immunosuppressive therapy who underwent serum galactomannan antigen testing for suspected IPA. Patients who fulfilled the clinical features or mycological evidence as per the EORTC/MSGERC definition were considered as having “potential IPA.” We compared the clinical characteristics of 364 patients who were categorized into 3 groups—potential IPA (n = 29), proven/probable IPA (n = 24), and non-IPA (n = 311; not meeting any definition). The potential and proven/probable IPA groups had significantly lower survival rates than the non-IPA group (*p* < 0.001). The potential IPA (adjusted hazard ratio [aHR], 2.0; 95% confidence interval [CI], 1.1–3.8) and proven/probable IPA (aHR, 2.6; 95% CI, 1.4–4.9) were independent risk factors for mortality. Compared with the EORTC/MSGERC definition, our proposed criteria improved sensitivity based on the diagnosis at the end of observation (50.0%, 100.0%, respectively). The characteristics and mortality rates of patients were similar between the potential and proven/probable IPA groups. Using these criteria for clinical diagnosis may provide high sensitivity.

## 1. Introduction

The advent of new immunosuppressive and biological agents has improved the prognosis of patients with AIIRDs over the past decade. However, opportunistic infections remain a major challenge [1]. Invasive pulmonary aspergillosis (IPA) is one of the most important opportunistic infections, and is the third most common invasive fungal disease (IFD), following pneumocystis pneumonia and candidemia, with a mortality rate of 25–70% [2,3]. Therefore, timely diagnosis and treatment are of paramount importance.

The European Organization for Research and Treatment of Cancer/Mycosis Study Group Education and Research Consortium (EORTC/MSGERC) consensus definition is the most widely used criteria worldwide for diagnosing IFDs, including IPA [4]. The EORTC/MSGERC definition considers three elements: host factors, clinical features, and mycological evidence. The EORTC/MSGERC definition was originally proposed for use in patients with hematologic malignancies in clinical trials [5]. Nonetheless, it is frequently used in clinical practice to guide management in various patients beyond the original target population, including those with AIIRDs [6]. As the sensitivity and specificity of the EORTC/MSGERC definition depend on the unique characteristics of each patient population, criteria have been proposed for research classification and/or clinical diagnosis in specific populations, such as patients in the intensive care unit [7] and those with chronic obstructive pulmonary disease (COPD) [8], influenza [9].

Considering their unique characteristics, patients with AIIRDs require special attention. Previous studies have repeatedly indicated that the EORTC/MSGERC definition has limited sensitivity as patients with AIIRDs often do not fulfill the host factor requirement. Many cases of IPA have been reported in patients with AIIRD who have only received low-dose corticosteroids or other immunosuppressive therapies [10,11,12]. Moreover, they do not often fulfill the clinical features’ requirement, such as typical radiologic findings [12,13,14,15]. The difficulty in diagnosing IPA among patients with AIIRDs is reflected by the fact that it is frequently first diagnosed postmortem [16]. There is concern that many cases of IPA in AIIRD patients may be overlooked. Therefore, it is essential to establish better criteria to identify patients who might benefit from antifungal treatment in a timely manner within a clinical setting [17,18]. Nonetheless, to the best of our knowledge, no criteria have been developed specifically for patients with AIIRDs.

Therefore, this study proposed criteria for patients with AIIRDs that can be applied in clinical settings and clinical trials by modifying the EORTC/MSGERC definition [19]. To prove the concept, we compared this criteria with the EORTC/MSGERC 2019 definition to explore the differences in clinical characteristics and prognosis for different categories.

## 2. Materials and Methods

### 2.1. Study Design and Participants

This single-center, retrospective, observational cohort study was performed at the Institute of Science of Tokyo Hospital (STH), Tokyo, Japan. It is an 813-bed, tertiary care academic hospital. The Department of Rheumatology conducts approximately 5000 outpatient visits and 300 hospitalizations annually. In the current study, we included patients who met the following criteria: visited the rheumatology clinic or required hospitalization in the Department of Rheumatology between June 2007 and July 2022; underwent serum galactomannan antigen (GMA) testing for suspected IPA; and received immunosuppressive therapy for AIIRDs at the time of GMA testing. To focus on IPA occurring in immunosuppressed patients with AIIRDs, the following exclusion criteria were considered: receipt of immunosuppressive therapy for conditions other than AIIRDs within 2 years of enrollment (e.g., antitumor chemotherapy); history of known IPA prior to the initiation of immunosuppressive therapy for AIIRDs; and serum GMA testing performed without a suspicion of new IPA (e.g., screening or follow-up).

Considering the retrospective nature of the study, the requirement for patient consent was waived. Our study complied with the principles of the Declaration of Helsinki, and the protocol was approved by the local ethics committee (13 November 2023, Permission number C2023-041). The reporting of this study complies with the Strengthening the Reporting of Observational Studies in Epidemiology guidelines.

### 2.2. Proposed Definition of IPA

Based on the EORTC/MSGERC 2019 definition [4], we developed criteria for diagnosing patients with AIIRDs by adding the category of “potential IPA.” Eligible patients in this study were categorized into proven/probable IPA, potential IPA, and non-IPA groups. The potential IPA group comprised patients who fulfilled the clinical features or mycological evidence of the EORTC/MSGERC 2019 definition and exhibited symptomatic pulmonary lesions that progressed over days to weeks, and other etiologies were dismissed (Table 1). The host factor requirement was also relaxed to include all forms of immunosuppressive therapy. All these criteria were intended to improve the sensitivity of clinical diagnosis by expanding “possible IPA.” *Aspergillus* PCR, which is included in the original EORTC/MSGERC definition, was not used to collect mycological evidence because it was not approved in Japan or available at STH. A comparison of our proposed criteria with the EORTC/MSGERC 2019 definition [4] and others for specific populations is presented in Appendix A.

The classifications of potential IPA, proven/probable IPA, and non-IPA were ascertained by two infectious disease (ID) physicians (TK and KO) based on a manual chart review. In cases of disagreement, consensus was reached through discussion with a third ID physician (NS). The classification was performed according to the information available within 2 weeks after conducting initial serum GMA testing for suspected IPA to taking into account the time for a diagnostic work-up. Similarly, the final diagnosis was performed using the information available at the end of the observation period.

### 2.3. Outcomes and Variables

The primary endpoint was survival calculated from the initial day of serum GMA testing. The observation was censored at the time of death or on 31 March 2024. A manual chart review was performed to collect clinical information. GMA and (1,3)-beta-d-glucan (BDG) levels were measured using the Platelia *Aspergillus* Ag assay (reference value, <0.5; Bio-Rad, Hercules, CA, USA) and Fungitec G-test MK-II (reference value, <20.0 pg/mL; Shimadzu, Kyoto, Japan), respectively.

### 2.4. Diagnostic Performance

The diagnostic performances of our proposed criteria and the EORTC/MSGERC 2019 definition were evaluated by estimating their sensitivity, specificity, positive predictive value, and negative predictive value, alongside their respective 95% CIs, with the final diagnosis serving as the gold standard.

### 2.5. Statistical Analysis

Categorical variables were expressed as n (%), and continuous variables were expressed as the median and interquartile range. Comparisons among each group were performed using one-way ANOVA for continuous variables and the chi-squared test for categorical variables. Survival curves were constructed using the Kaplan–Meier method, and the log-rank test was used to assess significant differences among the three groups. Cox proportional hazard regression analysis was performed to estimate the hazard ratios (HRs) and 95% CIs. Factors such as age and the presence of interstitial lung disease (ILD) were included in the multivariate model owing to their impact on mortality [20,21]. All *p*-values were two-tailed, and *p*-values of <0.05 were considered to indicate statistical significance. All analyses were performed using SAS version 9.4 (SAS Institute Inc., Cary, NC, USA).

## 3. Results

### 3.1. Study Population Characteristics

During the study period, 1324 patients underwent serum GMA testing in the Department of Rheumatology at STH (Figure 1). After applying the exclusion criteria and excluding patients without AIIRDs, 364 patients were analyzed. Overall, 24 (6.6%) patients had proven/probable IPA, 29 (7.9%) had potential IPA, and 311 (85.4%) had non-IPA.

### 3.2. Differences in Clinical Characteristics

The clinical characteristics of the patients in each group are presented in Table 2. Age (*p* = 0.023) and the proportion of male patients (*p* = 0.023) differed among the three groups. Neutrophil counts differed significantly among the three groups (*p* = 0.044), however few patients had neutrophil counts of <500/µL for >10 days (no patients with proven/probable IPA, one with potential IPA, and two with non-IPA). The incidences of systemic vasculitis as an underlying disease (*p* < 0.001) and ILD as a pulmonary complication (*p* = 0.004) differed among the three groups. In the proven/probable and potential IPA groups, the patterns of abnormal radiologic findings of the lungs were characterized by “dense, well-circumscribed lesions,” “air crescent,” and “cavity,” which corresponded to the clinical features described in the EORTC/MSGERC 2019 definition.

### 3.3. Survival Analysis

The survival curves of the proven/probable IPA, potential IPA, and non-IPA groups are presented in Figure 2. Patients with proven/probable IPA and those with potential IPA had significantly lower survival rates than those with non-IPA (*p* < 0.001). After adjusting for age and the presence of ILD using the Cox proportional hazards model, compared with the non-IPA group, the proven/probable IPA (adjusted hazard ratio [aHR] = 2.6; 95% CI = 1.4–4.9; *p* < 0.01) and potential IPA (aHR = 2.0; 95% CI = 1.1–3.8; *p* = 0.03) groups were independently associated with higher mortality (Table 3).

### 3.4. Final Diagnosis of Pulmonary Diseases and Antifungal Treatment

Overall, 4.2% (1/24) of patients with proven/probable IPA and 24.1% (7/29) of those with potential IPA had a final diagnosis other than IPA. Among 311 patients with non-IPA, the final diagnoses included other infections (n = 177, 57.1%), worsening of primary disease (n = 86, 23.4%), malignancy (n = 25, 6.8%), and drug-induced lung disease (n = 12, 3.3%; Table 4). Pulmonary disease remained undiagnosed in eight (2.6%) patients, only two of whom were treated for IPA by a primary rheumatologist. In contrast, 83.0% of patients with either proven/probable or potential IPA were prescribed antifungals by primary rheumatologists (Appendix A). The potential IPA group had a significantly higher proportion of culture-negative cases than the proven/probable IPA group (48.2% vs. 0.0%, *p* = 0.002, Appendix A).

### 3.5. Comparison of Our Proposed Criteria and the EORTC/MSGERC 2019 Definition

Table 5 presents the diagnostic performances of our proposed criteria and the EORTC/MSGERC 2019 definition. The diagnostic sensitivities of potential IPA defined by our proposed criteria, proven/probable IPA, and possible IPA defined by the EORTC/MSGERC 2019 definition were 100%, 50.0%, and 61.7%, respectively. The specificities of potential IPA, proven/probable IPA, and possible IPA were 97.8%, 99.7%, and 99.1%, respectively.

Figure 3 presents the missing factors for patients with potential IPA that would have led to a classification of probable IPA. The most common missing factor was clinical features in 14 patients, followed by host factors in 9 patients and mycological evidence in 7 patients. Among patients lacking mycological evidence, five met the possible IPA criteria in accordance with the EORTC/MSGERC 2019 definition. A schematic diagram of each classification group is presented in Appendix A.

## 4. Discussion

Patients with potential IPA, which was based on our newly proposed criteria specifically targeting patients with AIIRDs, had similar survival rates as those with proven/probable IPA but worse survival rates than those with non-IPA. In addition, our proposed criteria had statistically higher sensitivity than the EORTC/MSGERC 2019 definition while maintaining high specificity in diagnosing IPA.

The clinical characteristics of patients were extremely similar between the potential IPA and proven/probable IPA groups. Both groups rarely had neutropenia. The patients generally received high corticosteroid doses and showed high rates of systemic vasculitis and/or ILDs. These findings were consistent with the characteristics of IPA in patients with AIIRDs reported in previous observational studies [12,13,14,15,22]. We speculated that structural changes secondary to ILDs predispose patients to *Aspergillus* colonization, as observed in patients with COPD [8,23]. In contrast, the potential IPA group had lower rates of hemoptysis, fever, positive cultures, and typical radiographic findings than the proven/probable IPA group. Notably, these differences, excluding those related to clinical symptoms, were pertinent to the definition of potential IPA itself. As the criteria have been expanded with the intention of improving diagnostic sensitivity, we hypothesized that the potential IPA group includes heterogeneous patients, involving those with “true IPA” and “airway colonization” [24]. The differences in the clinical symptoms between these groups might reflect this heterogeneity.

Most patients with potential IPA in this cohort did not meet the criteria for possible IPA, as stipulated by the EORTC/MSGERC 2019 definition. Nonetheless, the survival rate was approximately 50% in the proven/probable and potential IPA groups, in line with previous findings on IPA among patients with AIIRDs [2,3]. The difference remained even after adjusting for age and the presence of ILDs. Furthermore, the causes of mortality were similar between the proven/probable and potential IPA groups. Together with the other clinical characteristics, we hypothesized that patients with potential IPA may benefit from antifungal therapy.

In our cohort, the addition of potential IPA in our criteria greatly increased the diagnostic sensitivity without significantly compromising specificity compared with the EORTC/MSGERC 2019 definition. This increase was attributable to the expansion of host factors and clinical features. Since patients with AIIRDs can develop IPA at lower steroid doses, other risk factors, such as the presence of ILD, might play an important role in the development of IPA. Clinical features in the EORTC/MSGERC 2019 definition, such as halo signs and cavities, reflect angioinvasion by hyphae, which commonly occurs in patients with neutropenia [25,26], but rarely patients with AIIRDs [12,13,14,15]. We speculate that these characteristics contribute to false-negative results when using the EORTC/MSGERC 2019 definition.

It remains unexplored whether early antifungal therapy improves the prognosis of IPA in patients with AIIRDs. In this study, most patients with IPA received antifungal agents, making it difficult to compare patients with and without treatment. Although early antifungal therapy for proven/probable IPA has been reported to improve prognosis in patients with severe neutropenia [27], its efficacy in patients without neutropenia, including those with AIIRDs, remains unknown. One of the major reasons was that enrolling patients with AIIRDs in clinical trials is challenging given the lack of an appropriate early diagnosis [18], in contrast to patients with hematological malignancies who can be enrolled in clinical trials using the EORTC/MSGERC 2019 definition [28,29]. In the future, our proposed criteria are expected to facilitate the conduct of clinical trials, including those aimed at evaluating the efficacy of early antifungal therapies targeting IPA among patients with AIIRDs.

We acknowledge that our study had limitations. First, owing to its single-center retrospective design, our study results had information bias and limited generalizability. Pathological examination was rarely performed, and the final diagnosis relied on the clinical information documented in the medical records. Although the three ID physicians independently reviewed the cases to maximize the certainty, this remains a major limitation. In addition, bronchoalveolar lavage, which is a highly sensitive method for diagnosis [30,31], was not sufficiently performed in this cohort, potentially leading to an underdiagnosis of IPA. Also, the sample size of patients with IPA was insufficient to allow complete adjustment for potential confounders. Therefore, our results do not have sufficient external validity to enable extrapolation to the daily clinical practice among patients with AIIRDs. Meanwhile, the criteria development methodology in our study was not based on a systematic review or consensus within an expert working group, but on the insights from previous reports [17,18]. However, this study might encourage future validation, which would lead to improved outcomes for patients with AIIRDs through early diagnosis and treatment.

In conclusion, patients with potential IPA identified using our proposed criteria had similar clinical characteristics and mortality rates as those with proven/probable IPA identified by the EORTC/MSGERC 2019 definition. Using these criteria for clinical diagnosis may provide higher sensitivity compared with the EORTC/MSGERC definition. Further validation is needed to determine whether the use of these criteria can improve the prognosis of such patients.

## Figures and Tables

**Figure 1 jof-11-00437-f001:**
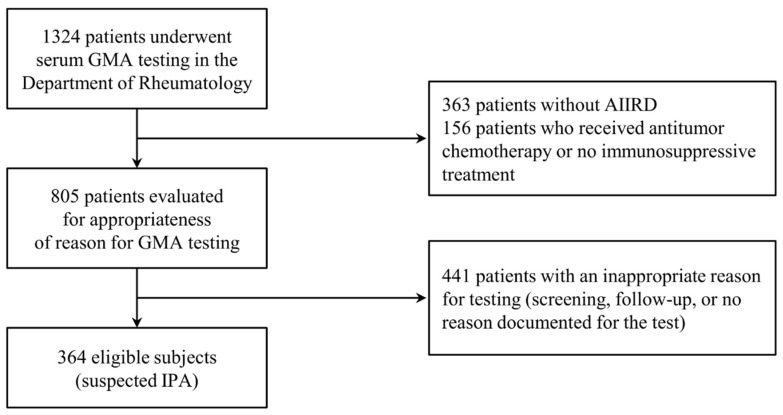
Flow diagram of patient enrollment in this study. Abbreviations: AIIRDs, autoimmune inflammatory rheumatic diseases; GMA, galactomannan antigen; IPA, invasive pulmonary aspergillosis.

**Figure 2 jof-11-00437-f002:**
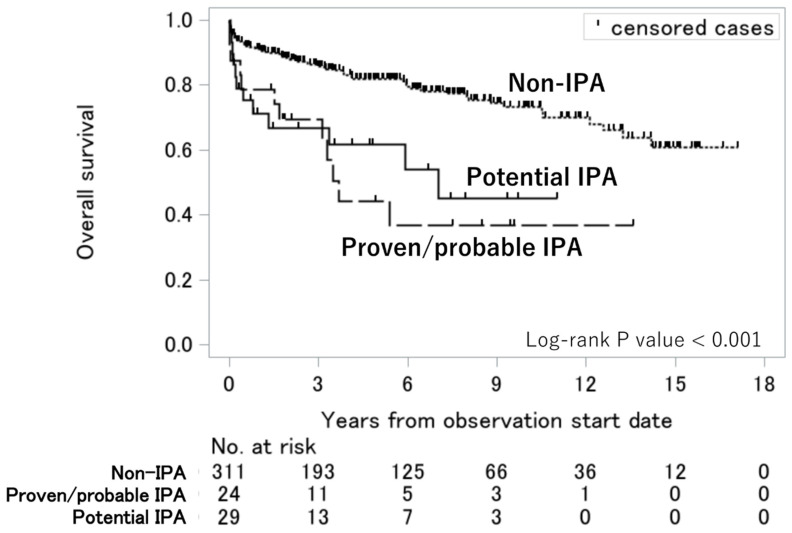
Overall survival among the proven/probable IPA, potential IPA, and non-IPA groups. Survival curves were constructed using the Kaplan–Meier method, and the log-rank test was used to assess significant differences among the three groups. Abbreviation: IPA, invasive pulmonary aspergillosis.

**Figure 3 jof-11-00437-f003:**
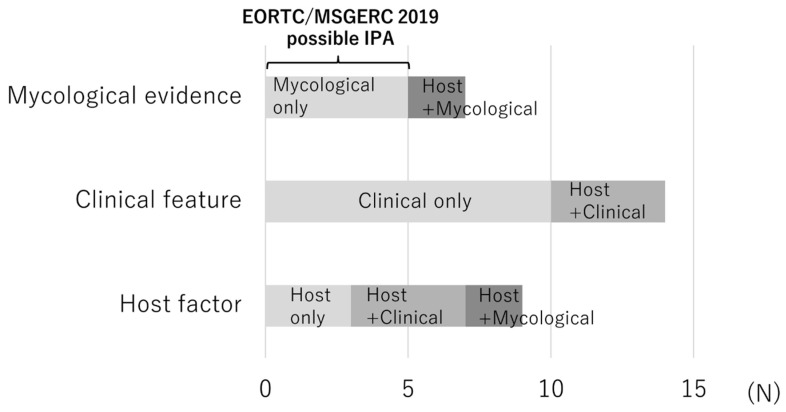
Reasons for non-classification of potential IPA as probable IPA by the EORTC/MSGERC 2019 definition (missing factors) for each case. Each bar indicates the missing factors among the three components of the EORTC/MSGERC 2019 definition that would have allowed the case of potential IPA to be classified as probable IPA. Abbreviations: EORTC/MSGERC, European Organization for Research and Treatment of Cancer/Mycosis Study Group Education and Research Consortium; IPA, invasive pulmonary aspergillosis.

**Table 1 jof-11-00437-t001:** Definition of potential IPA in this study.

Potential IPA
Required: All of the following three criteria are fulfilled.
New pulmonary lesions with any respiratory symptoms progressing on a daily to weekly time course with no response to antimicrobial or immunosuppressive therapy.
2. No obvious evidence of other diseases (e.g., worsening of primary disease, malignancy).
3. Either of the following findings exist.
The presence of one of the following four patterns on CT (*1): Dense, well-circumscribed lesions with or without a halo sign Air crescent sign Cavity Wedge-shaped and segmental or lobar consolidation
The presence of one of the following findings (*2): Microscopical detection of fungal elements in the sputum, BAL, bronchial brush, or aspirate indicating the presence of *Aspergillus* species *Aspergillus* species recovered by culture from the sputum, BAL, bronchial brush, or aspirate GMA detected in a single serum or plasma sample (≥1.0) GMA detected in the BAL fluid (≥1.0) GMA detected in a single serum or plasma sample (≥0.7) and BAL fluid (≥0.8)

(*1) This item indicates fulfilling the “clinical features” of pulmonary aspergillosis in the EORTC/MSGERC 2019 definition (i.e., almost equal to the EORTC/MSGERC 2019 “possible IPA” category). (*2) This item indicates fulfilling the “mycological evidence” of pulmonary aspergillosis in the EORTC/MSGERC 2019 definition. Abbreviations: BAL, bronchoalveolar lavage; CT, computed tomography; IPA, invasive pulmonary aspergillosis; EORTC/MSGERC, European Organization for Research and Treatment of Cancer/Mycosis Study Group Education and Research Consortium; GMA, galactomannan antigen.

**Table 2 jof-11-00437-t002:** Characteristics of each category of patients with AIIRDs.

	All Patients	Proven/Probable IPA	Potential IPA	Non-IPA	*p*
Total number	364 (100%)	24 (100%)	29 (100%)	311 (100%)	-
Age at enrollment (years)	69 (61–76)	70.5 (64.5–74.5)	73 (68–77)	68 (60–75)	0.023 *
Follow-up period (years)	4.3 (1.5–7.9)	2.0 (0.9–5.2)	1.5 (0.3–5.9)	4.8 (1.8–8.2)	0.007 *
Sex (male)	148 (40.7%)	17 (70.8%)	16 (55.2%)	115 (37.0%)	0.001 *
Diabetes	146 (40.3%)	11 (45.8%)	14 (48.3%)	121 (39.2%)	0.538
CKD	134 (36.8%)	10 (41.7%)	11 (37.9%)	113 (36.3%)	0.865
Smoking history	156 (44.1%)	16 (66.7%)	12 (42.9%)	128 (42.4%)	0.069
Underlying AIIRD					
RA	180 (49.5%)	11 (45.8%)	11 (37.9%)	158 (50.8%)	0.388
SLE	24 (6.6%)	1 (4.2%)	1 (3.4%)	22 (7.1%)	0.666
Inflammatory myositis	45 (12.4%)	2 (8.3%)	3 (10.3%)	40 (12.9%)	0.763
Systemic vasculitis	68 (18.7%)	8 (33.3%)	13 (44.8%)	47 (15.1%)	<0.001 *
Systemic sclerosis	22 (6.0%)	0 (0.0%)	0 (0.0%)	22 (7.1%)	0.136
Other AIIRD	8 (2.2%)	0 (0.0%)	0 (0.0%)	8 (2.6%)	0.498
Background lung disease (repeatable)					
ILD	157 (43.1%)	15 (62.5%)	19 (65.5%)	123 (39.5%)	0.004 *
Honeycomb lung	35 (9.6%)	8 (33.3%)	4 (13.8%)	23 (7.4%)	<0.001 *
COPD	34 (9.3%)	2 (8.3%)	4 (13.8%)	28 (9.0%)	0.687
Bronchiectasis	48 (13.2%)	6 (25.0%)	7 (24.1%)	35 (11.3%)	0.031 *
NTM	19 (5.2%)	0 (0.0%)	0 (0.0%)	19 (6.1%)	0.181
Old tuberculosis	20 (5.5%)	1 (4.2%)	2 (6.9%)	17 (5.5%)	0.909
Immunosuppressive agent (repeatable)					
PSL (dose, mg/day)	5 (1.5–15.0)	13.8 (4.0–30.0)	6.0 (4.0–20.0)	5.0 (1.0–13.0)	<0.001 *
≥10 mg/day	135 (37.1%)	15 (62.5%)	10 (34.5%)	110 (35.4%)	0.028 *
≥20 mg/day	68 (18.7%)	10 (41.7%)	8 (27.6%)	50 (16.1%)	0.004 *
≥30 mg/day	38 (10.4%)	6 (25.0%)	5 (17.2%)	27 (8.7%)	0.019 *
Methotrexate	104 (28.6%)	3 (12.5%)	3 (10.3%)	98 (31.5%)	0.011 *
Calcineurin inhibitor	91 (25.0%)	6 (25.0%)	9 (31.0%)	76 (24.4%)	0.735
Cyclophosphamide	33 (9.1%)	4 (16.7%)	3 (10.3%)	26 (8.4%)	0.381
Azathioprine	26 (7.1%)	3 (12.5%)	3 (10.3%)	20 (6.4%)	0.422
TNF inhibitor	45 (12.4%)	4 (16.7%)	0 (0.0%)	41 (13.2%)	0.096
IL-6 inhibitor	18 (4.9%)	1 (4.2%)	1 (3.4%)	16 (5.1%)	0.907
Abatacept	14 (3.8%)	1 (4.2%)	0 (0.0%)	13 (4.2%)	0.532
JAK inhibitor	3 (0.8%)	0 (0.0%)	1 (3.4%)	2 (0.6%)	0.251
Mycophenolate mofetil	9 (2.5%)	1 (4.2%)	0 (0.0%)	8 (2.6%)	0.596
Rituximab	8 (2.2%)	1 (4.2%)	0 (0.0%)	7 (2.3%)	0.580
Clinical symptoms (repeatable)					
Fever	148 (40.8%)	10 (41.7%)	4 (13.8%)	134 (43.2%)	0.009 *
Dyspnea	73 (20.1%)	4 (16.7%)	3 (10.3%)	66 (21.3%)	0.338
Hemoptysis	36 (9.9%)	10 (41.7%)	3 (10.3%)	23 (7.4%)	<0.001 *
Sputum	168 (46.3%)	13 (54.2%)	9 (31.0%)	146 (47.1%)	0.191
Cough	201 (55.4%)	14 (58.3%)	17 (58.6%)	170 (54.8%)	0.885
Chest pain	22 (6.1%)	0 (0.0%)	0 (0.0%)	22 (7.1%)	0.135
Fatigue	90 (24.8%)	6 (25.0%)	1 (3.4%)	83 (26.8%)	0.021
Patterns of abnormal shadows (repeatable)					
Bilateral shadows	215 (60.6%)	12 (50.0%)	17 (60.7%)	186 (61.4%)	0.547
Multiple shadows	107 (30.1%)	14 (58.3%)	10 (35.7%)	83 (27.4%)	0.005 *
Nodule	140 (39.4%)	18 (75.0%)	17 (60.7%)	105 (34.7%)	<0.001 *
Dense, well-circumscribed lesions	30 (8.5%)	9 (37.5%)	8 (28.6%)	13 (4.3%)	<0.001 *
Air crescent sign	7 (2.0%)	4 (16.7%)	3 (10.7%)	0 (0.0%)	<0.001 *
Cavity	53 (14.9%)	15 (62.5%)	10 (35.7%)	28 (9.2%)	<0.001 *
Wedge-shaped and segmental or lobar consolidation	73 (20.6%)	11 (45.8%)	3 (10.7%)	59 (19.5%)	0.004 *
Pleural effusion	39 (11.0%)	4 (16.7%)	3 (10.7%)	32 (10.6%)	0.654
Clinical laboratory tests					
Neutrophil count (/µL)	6310 (4074–9021)	8477 (6540–10,917)	6177 (4606–8131)	6037 (3915–8733)	0.044 *
Serum BDG (pg/mL)	0.0 (0.0–66.7)	14.1 (0.0–66.7)	0.0 (0.0–26.4)	0.0 (0.0–0.0)	<0.001 *
Serum GMA	0.2 (0.1–0.4)	0.3 (0.2–1.0)	0.7 (0.2–1.3)	0.2 (0.1–0.3)	<0.001 *
<0.5 (N)	282 (77.7%)	15 (62.5%)	12 (41.4%)	255 (82.3%)	<0.001 *
0.5–0.9 (N)	47 (12.9%)	3 (12.5%)	7 (24.1%)	37 (11.9%)	
≥1.0 (N)	34 (9.4%)	6 (25.0%)	10 (34.5%)	18 (5.8%)	
BAL					
Number of tests	121 (33.2%)	13 (54.2%)	11 (37.9%)	97 (31.2%)	0.060
Culture positivity (*Aspergillus* sp.)	16 (4.4%)	7 (29.2%)	4 (13.8%)	5 (1.6%)	<0.001 *
Available BALF-GMA	2 (0.5%)	0 (0.0%)	1 (3.4%)	1 (0.3%)	0.087
BALF-GMA positivity (≥0.8)	0 (0.0%)	0 (0.0%)	0 (0.0%)	0 (0.0%)	-

Categorical variables were presented as n (%) unless otherwise indicated. Continuous variables were presented as the median (interquartile range). *p*-values were calculated for comparisons among the three groups (proven/probable IPA, potential IPA, and non-IPA) using one-way ANOVA for continuous variables and the chi-squared test for categorical variables. * *p*-values of <0.05 were considered to indicate statistical significance. Abbreviations: AIIRDs, autoimmune inflammatory rheumatic diseases; BAL, bronchoalveolar lavage; BALF, bronchoalveolar lavage fluid; BDG,(1,3)-beta-D-glucan; CKD, chronic kidney disease; COPD, chronic obstructive pulmonary disease; GMA, galactomannan antigen; IPA, invasive pulmonary aspergillosis; IL-6, interleukin-6; JAK, Janus kinase; ILD, interstitial lung disease; N, number; NTM, nontuberculous mycobacteria; PSL, prednisolone; RA, rheumatoid arthritis; TNF, tumor necrosis factor; SLE, systemic lupus erythematosus.

**Table 3 jof-11-00437-t003:** Factors associated with mortality and their HRs.

	Adjusted HR (95% CI)	*p*
Proven/probable IPA	2.630 (1.404–4.929)	0.003 *
Potential IPA	2.017 (1.067–3.815)	0.031 *
Age (≥65 years)	2.216 (1.422–3.453)	<0.001 *
Interstitial lung disease	1.560 (0.962–2.530)	0.071

The adjusted HRs and 95% CIs of factors contributing to mortality were estimated using multivariate Cox proportional hazard regression analysis. * *p*-values of <0.05 were considered to indicate statistical significance. Abbreviations: IPA, invasive pulmonary aspergillosis; HR, hazard ratio; CI, confidence interval.

**Table 4 jof-11-00437-t004:** Final diagnosis of pulmonary lesions by manual chart review in each category.

	All Patients	Proven/Probable IPA	Potential IPA	Non-IPA
Total number	364 (100%)	24 (100%)	29 (100%)	311 (100%)
IPA	47 (12.9%)	23 (95.8%)	24 (80.0%)	0 (0.0%)
Other pulmonary infections	178 (48.9%)	0 (0.0%)	1 (3.3%)	177 (57.1%)
Bacterial	83 (22.8%)	0 (0.0%)	0 (0.0%)	83 (26.8%)
NTM	34 (9.3%)	0 (0.0%)	1 (3.3%)	33 (10.6%)
PCP	30 (8.2%)	0 (0.0%)	0 (0.0%)	30 (9.7%)
Viral	15 (4.1%)	0 (0.0%)	0 (0.0%)	15 (4.8%)
Nocardiosis	11 (3.0%)	0 (0.0%)	0 (0.0%)	11 (3.5%)
Tuberculosis	5 (1.4%)	0 (0.0%)	0 (0.0%)	5 (1.6%)
Miscellaneous	4 (1.1%)	0 (0.0%)	0 (0.0%)	4 (1.3%)
Attributable to the primary disease	83 (22.8%)	1 (4.2%)	3 (10.0%)	79 (25.5%)
Malignancy	25 (6.9%)	0 (0.0%)	1 (3.3%)	24 (7.7%)
Drug-induced lung disease	12 (3.3%)	0 (0.0%)	0 (0.0%)	12 (3.9%)
Unable to diagnose	10 (2.7%)	0 (0.0%)	1 (3.3%)	9 (2.9%)
Other	6 (1.6%)	0 (0.0%)	0 (0.0%)	6 (1.9%)

Categorical variables were presented as n (%) unless otherwise indicated. All diagnoses were mutually exclusive. “Miscellaneous” included two cases of sinus fungal infections, one case of pulmonary cryptococcosis, and one case of allergic bronchopulmonary aspergillosis. “Malignancy” included 13 cases of primary lung carcinomas, 10 cases of methotrexate-related lymphoproliferative diseases, 1 case of metastatic lung tumor, and 1 case of cancer of unknown primary. “Other” included two cases of pulmonary embolism, one case of chronic bronchitis, one case of bronchial asthma, and one case of hypersensitivity pneumonitis. Abbreviations: IPA, invasive pulmonary aspergillosis; NTM, nontuberculous mycobacteria; PCP, pneumocystis pneumonia.

**Table 5 jof-11-00437-t005:** Diagnostic performance of our proposed criteria with the category of “potential IPA” versus the EORTC/MSGERC 2019 definition.

	Sensitivity, % (95% CI)	Specificity, % (95% CI)	PPV, % (95% CI)	NPV, % (95% CI)
Our criteria				
Proven + probable + potential	100.0 (100.0–100.0)	97.8 (96.2–99.4)	86.8 (77.7–95.9)	100.0 (100.0–100.0)
EORTC/MSGERC 2019				
Proven only	6.5 (<0.1–13.7)	100.0 (100.0–100.0)	100.0 (100.0–100.0)	88.1 (84.7–91.4)
Proven + probable	50.0 (35.6–64.4)	99.7 (99.1–100.0)	95.8 (87.8–100.0)	93.2 (90.6–95.9)
Proven + probable + possible	61.7 (48.6–75.0)	99.1 (98.0–100.0)	90.3 (79.9–100.0)	94.6 (92.2–97.0)

We calculated the sensitivity, specificity, PPV, and NPV for each diagnostic category using the final diagnosis of IPA as the gold standard. Abbreviations: EORTC/MSGERC, European Organization for Research and Treatment of Cancer/Mycosis Study Group Education and Research Consortium; IPA, invasive pulmonary aspergillosis; NPV, negative predictive value; PPV, positive predictive value; CI, confidence interval.

## Data Availability

The original contributions presented in this study are included in the article/Appendix A. Further inquiries can be directed to the corresponding author.

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
