# Peer review of "Proposed Diagnostic Criteria for Invasive Pulmonary Aspergillosis in Patients with Autoimmune Inflammatory Rheumatic Diseases: A Proof-of-Concept Study [Author-notes fn1-jof-11-00437]"

_jof, 2025, doi:10.3390/jof11060437_

Round 1
Reviewer 1 Report
I have read this interesting study that shows that partial adherence to the EORTC/MSGERC definition can improve diagnostic sensitivity. This is a highly important study in the diagnosis of invasive pulmonary aspergillosis in patients with rheumatic diseases. As the authors point out, despite the results obtained, the use of these criteria in patient prognosis needs to be validated. The study is scientifically rigorous, well-presented, and clearly written.
I only have two minor suggestions:
Please, enter the date of approval by the institutional ethics committee (Line 88).
Table 3: To make it more readable, please include the % symbol in the headings of each column, not in each number.

Reviewer 2 Report
Dear Authors;
Thank you very much for this article.
Undoubtedly, this represents a contribution to the diagnosis of patients with AIIRDs.
Several corrections and suggestions are included in the PDF file.
There are several inaccuracies or incomplete data.
Table 3 seems confusing and inaccurate. For example, it would be essential to know the characteristics associated with those 13 deaths: what therapy was used in those patients? What species was identified in them? Was a GM performed on them? And in the others? Was B-D glucan performed?
Are the radiological data of the patients shown in the same table available?
Kind regards
Table 3 seems confusing and inaccurate. For example, it would be essential to know the characteristics associated with those 13 deaths: what therapy was used in those patients? What species was identified in them? Was a GM performed on them? And in the others? Was B-D glucan performed?
Are the radiological data of the patients shown in the same table available?
Table 2 is confusing, maybe you can link the two tables, or some of the info contained in table 2 can be eliminated. In example who patients die? which AF received?
